# Analysis of the Air-Reversed Brayton Heat Pump with Different Layouts of Turbochargers for Space Heating

**Shugang Wang [1], Shuangshuang Li [2,*], Shuang Jiang [3]** and **Xiaozhou Wu [1]**

1 Faculty of Infrastructure Engineering, Dalian University of Technology, Dalian 116024, China;
sgwang@dlut.edu.cn (S.W.); fonen519@dlut.edu.cn (X.W.)
2 College of Civil Engineering and Architecture, Dalian University, Dalian 116622, China
3 College of Civil Engineering, Dalian Minzu University, Dalian 116600, China; shjiang@dlnu.edu.cn
* Correspondence: lishuangshuang@dlu.edu.cn; Tel.: +86-155-4268-6558

**Abstract:** The air-reversed Brayton cycle produces charming, environmentally friendly effects by using air as its refrigerant and has potential energy efficiency in applications related to space heating and building heating. However, there exist several types of cycle that need to be discussed. In this paper, six types of air-reversed Brayton heat pump with a turbocharger, applicable under different heating conditions, are developed. The expressions of the heating coefficient of performance (COP) and the corresponding turbine pressure ratio are derived based on thermodynamic analysis. By using these expressions, the effects of turbine pressure ratio on the COP under different working conditions are theoretically analyzed, and the optimal COPs of different cycles under specific working conditions are determined. It is observed that Cycles A and C have the highest heating COPs, and there is an optimal pressure ratio for each cycle. The corresponding pressure ratio of the optimal COP is different, concentrated in the range of 1.5–1.9. When the pressure ratio reaches the optimal value, increasing the pressure ratio does not significantly improve the heating COP. Take Cycle F as an example: the maximum error between the calculated results and experimental observation is lower than 5.6%. These results will enable further study of the air-reversed Brayton heat pump with a turbocharger from a different perspective.

**Keywords:** air-reversed Brayton heat pump; turbocharger; system layout; pressure ratio; COP

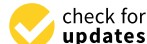



## 1. Introduction

The use of heat pumps for space heating and drying is one of the solutions for providing a stable and affordable energy supply which contributes to environmental protection and sustainable energy development. Although the theoretical coefficient of performance (COP) value is relatively low, air-source devices may have great potential for development. A reversed Brayton cycle using air as the green working fluid is a potential alternative to conventional vapor compression systems of heating and drying [1–3]. Recent exploitation of the unification of expansion/compression into a single operation (compander) promotes the productization of air-reversed Brayton heat pump (air cycle heat pump) products [4,5].

Air as the working fluid in the reversed Brayton cycle is a potential substitute for the conventional vapor compression cycles. However, compared with conventional vapor compression heat pumps, the energy efficiency of a basic air cycle heat pump (air-reversed Brayton heat pump) is lower under normal temperature conditions. Thus, a regenerated air cycle was utilized for improving the overall system performance. The performance of an air cycle heat pump can be remarkably enhanced by adding a regenerator [1,6,7]. Based on finite-time thermodynamics, an analytical solution was derived that demonstrated the effects of pressure ratio, heat exchanger effectiveness, and the ratio between the inlet temperature of the cooling fluid and that of the heating fluid of the heat reservoir on key performance indices [8–12].

The reversed Brayton cycle, which uses air as a refrigerant, has more typical operation characteristics compared with the traditional ones. Most notably, the regenerated air-reversed Brayton heat pump is more suitable for application in cold and frigid regions. It can not only break through the limits of the ambient temperature of air-source heat pumps but can also remove the heat exchange of the low-temperature side when using a semi-open form, which means that the air can directly enter the compressor and avoid various problems caused by evaporator frosting. A thermodynamic model of an air cycle heat pump with a compressor and an expander was proposed by Zhang et al.; heating capacity in line with heating load was found [13]. This thermodynamic model was further developed by Yuan and Zhang, who added a regenerator so that the regenerated air cycle heat pump could not only ensure heating capacity that was in line with the heating load but could also achieve a higher COP compared with trans-critical $CO_2$ heat pumps under a large temperature difference [14]. A simulation model of an air cycle heat pump water heater was developed by Yang et al., and it was found that this system could save the heating-up period, especially when operating at low ambient temperature conditions [15]. As the expansion device in the market is too large for domestic and vehicle heating usage, the existing experimental studies were primarily limited to improving the structure of individual components such as the compressor and the expander [16–18].

A turbocharger was used in air cycle systems to substitute the compressor and the expander. It miniaturized the air cycle system and also simplified the system design without considering the matching and connection problems between the compressor and the expander. A comprehensive report on an air cycle with a turbocharger was released by TNO [19], in which it was mentioned that a pilot plant processed air in open and recuperated cycles for freezing applications. Spence et al. [20,21] designed and established a prototype for existing trailer refrigeration units for road transport, where the overall COP was about 0.3 at the cryogenic temperature of −20 °C. Catalano et al. [22,23] designed an air cycle using a turbocharger and roots blowers for refrigeration. A COP higher than 0.6 was reported for the turbine exit temperature of −42 °C. Li et al. [24–26] theoretically and experimentally studied the air-reversed Brayton heat pump system with an automotive turbocharger. A thermodynamic model for this system was presented, and the relationships between the system performance and the operating parameters were illustrated. Moreover, a test bench of a regenerated air cycle heat pump with a turbocharger was presented, and the measurement results illustrated that the heating capacity of the air cycle heat pump could match the heating load well.

However, the form of the air-reversed Brayton cycle with a turbocharger in the current research is single, and the application scope is narrow. As the functions and application conditions of the single-cycle form are limited, it is necessary to derive a variety of cycle processes to improve this type of heat pump cycle form, and the difference in cycle form can also lead to changes in heating characteristics. This paper develops a different thermodynamic process for the air-reversed Brayton cycle with a turbocharger. The relations between the heating COPs and the turbine pressure ratio are established based on reasonable assumptions and the thermodynamic diagram. Therefore, an in-depth look into the variation of performance parameters with pressure ratio parameters in different cycles is achieved.

## 2. Theoretical Analysis of Different Types of Air Heat Pump with a Turbocharger

Compared with a traditional air-source heat pump system, the air-reversed Brayton heat pump system with a turbocharger has great potential in low-temperature heating, energy saving, and emission reduction. As the turbocharger is used to replace the compressor and expander, and the blower is added as the power source, it is possible to use the air-reversed Brayton heat pump system in civil building winter heating and domestic hot water supplies and other projects. By designing various thermodynamic processes of the air-reversed Brayton heat pump system with a turbocharger driven by a blower, the service

conditions and application scope of this kind of heat pump can be widened, which endows this system with greater diversity.

### 2.1. Description of Different Cycle Structures and Characteristics

Considering the limitations of heating objective conditions and the diversity of user needs, a single form of air-reversed Brayton heat pump with a turbocharger cannot meet all the application conditions. Furthermore, a variety of cycle forms have their own heating characteristics and structural advantages. Therefore, it is necessary to design more structural forms by designing combined modes of blowers and turbochargers, the number of blowers and their setting positions, the number of heat exchangers in the system, and their settings in the system position, etc. These forms can be extended from the six kinds of new heat pump cycle with a turbocharger to different heat demands and conditions, named Cycle A–Cycle F and shown in Figure 1. These six forms can not only provide different terminal heat-carrying fluids, such as water or air for heating and domestic hot water, but can also discharge cooler air from the system that can be used for refrigeration. In this section, heating is used as an example. Although these six kinds of cycle have certain representativeness, the structure is not limited to these six forms. Its application range can be further expanded by deriving other structures not covered in this paper.

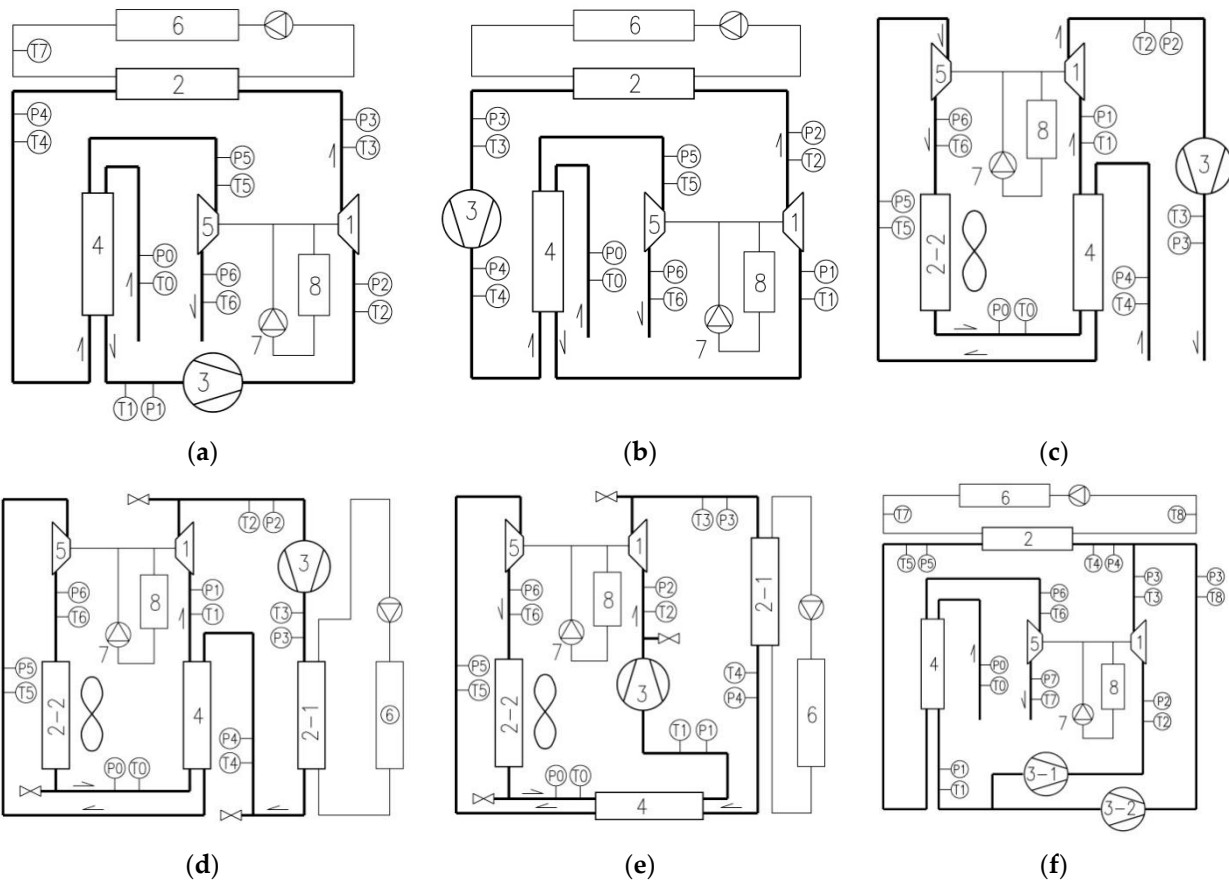

**Figure 1.** Different cycles of the air-reversed Brayton heat pump system with a turbocharger. 1: compressor, 2: heat exchanger, 3: blower, 4: preheater (regenerative), 5: turbine, 6: fan coil, 7: oil pump, 8: oil tank (P/T pressure sensor/temperature sensor). (**a**) Cycle A; (**b**) Cycle B; (**c**) Cycle C; (**d**) Cycle D; (**e**) Cycle E; (**f**) Cycle F.

Except for Cycle F, which was set as having two blowers, all other cycles adopt one blower as the power-driving equipment, i.e., a single-fan system. In Cycle A, the blower is set before the compressor of the turbocharger, and the cold air in the environment is initially preheated through preheater 4. This equipment can significantly improve the

heating performance of the system. Subsequently, the air first passes through the blower 3 and then enters the turbocharger compressor 1 to be heated up and boosted. The setting of oil pump 7 and oil tank 8 ensures the normal, high-speed operation of the turbocharger. High-temperature air through heat exchanger 2 is mainly used for the heat exchanger. The fan coil unit 6 and similar equipment can be used for the heating end equipment. The exothermic air still has some residual heat that is used as a heat source for preheater 4 to preheat the air from the environment. Finally, the air is expanded through the turbocharge turbine 5 and then diverted outside. In another semi-open cycle, Cycle B, the blower is placed in front of the turbine as shown in Figure 1b. The starting resistance is smaller than in Cycle A. These two forms are the most basic modifications of the traditional air-reversed Brayton heat pump cycle.

Cycle C is also a series combination of a single fan and a compressor. The indoor air first enters preheater 4 and then expands and cools through turbocharger turbine 5. Subsequently, it absorbs heat from the outdoor air heat source through heat exchanger 2. After heating up through preheater 4, it enters turbocharger compressor 1 and blower 3 and is finally sent into the room directly. The simple system structure can improve the temperature of the air entering the compressor and reduce the temperature of the air entering the turbine, which improves the heating efficiency of the system. Considering the practical application, blower 3 is set after the turbocharger compressor 1. This configuration can lower the exhaust pressure and temperature of the compressor, which is convenient for the recovery of lubricating oil at the outlet of the compressor. The hot air is directly fed into the room to reduce the heat loss caused by the end equipment. At the same time, system requirements are reduced as the system operates under a low vacuum.

Cycles D and E are closed circulation cycles which are not affected by the ambient air quality. However, due to the use of two heat exchangers, the overall efficiency of the whole machine is reduced, and the cost is higher than that of Cycles A and B. The above cycles have the same flow rates of the compressor and the turbine. By adding another blower, the system can be transformed into a dual-blower system, which is closer to the actual operating condition of the vehicle turbocharger. The circulation form of the dual-blower system, shown by Cycle F in Figure 1, is slightly more complicated than that of the single-blower system. One blower is connected in series with the compressor, which is subsequently in parallel with another blower. The air flows through the two parallel pipelines and is mixed together and then sent to gas-water heat exchanger 2 for heat transfer. Subsequently, the air is used to preheat the ambient air temperature and is finally discharged outdoors.

The heat-carrying fluid in the end equipment of the other cycles is water, except for in Cycle C. The equipment is not only limited to the fan coil, but can also use the floor radiant coil, radiator, etc., and is designed for a domestic hot water system. However, the temperature of the hot water is affected to a certain extent. To ensure the optimal pressure ratio of Cycle C, the air temperature as the circulating refrigerant is lower because the closed heat exchanger must be used on the heat-source side.

*2.2. Analytical Expression of Heating COP*

2.2.1. T-s Diagram

At present, only a limited amount of research exists on the air-reversed Brayton heat pump system with a turbocharger and a blower. Theoretical analysis is mostly carried out for the conventional, regenerated air-reversed Brayton heat pump system. However, this cannot truly reflect the heating characteristics of this system driven by a blower with a turbocharger and the relationship between the related parameters and the heating COP. Therefore, this section derives the expressions reflecting the analytical relation between the COP and the pressure ratio based on the temperature entropy diagram (*T-s* diagram) of different heat pump forms and relevant assumptions.

Figure 2 depicts the *T-s* diagram of each cycle shown in Figure 1, where the numbers correspond to the position of the temperature/pressure sensor (T/P) in Figure 1. Curve 1–2 in Cycle A represents the compression process of the blower; curves 2–3 and 5–6 represent

the compression and expansion processes in the turbocharger, respectively. Curve 3–4 represents the heat release process of the heat exchanger; curves 4–5 and 0–1 represent the heat release and heat absorption of the preheater, respectively.

Curves 1–2 and 5–6 in Cycle B represent the compression and expansion processes in the turbocharger, respectively. Curve 2–3 represents the heat release process of the heat exchanger; curve 3–4 indicates the compression process of the blower; curves 4–5 and 0–1 represent the heat release and heat absorption of the preheater, respectively. Curve 0–2 in Cycle C represents the compression process of the blower, and the hot air is directly sent to the end users; curves 1–2 and 4–5 represent the compression and expansion processes in the turbocharger, respectively. Curve 5–6 represents the heat release process of the heat exchanger; curves 3–4 and 6–1 represent the heat release and heat absorption processes of the regenerator, respectively.

Cycles D and E are closed cycles. Taking Cycle E as an example, curves 2–3 and 5–6 represent the compression and expansion processes in the turbocharger, respectively. Curve 1–2 represents the compression process of the blower; curves 4–5 and 0–1 represent the heat release and heat absorption process of the regenerator, respectively. Curve 6–0 represents the heat absorption process of the cold-end heat exchanger, the purpose of which is to obtain heat from the heat source. Such a system form is suitable as the system cannot use the open type, which is conducive to ensuring that the heat pump cycle working medium is kept clean and has low water content.

Cycle F is a semi-open cycle with two blowers. Curve 1–2 represents the compression process of the blower; curve 2–3 is the turbocharger compressor compression process; curve 1–8 represents the compression process of the parallel fan. Subsequently, point 8 is mixed with points 3–4; curve 4–5 represents the heat release process of the heat exchanger; curves 5–6 and 0–1 represent the heat release and heat absorption process of the preheater, respectively; curve 6–7 represents the expansion of the turbine in the turbocharger. The *T-s* diagram based on these different forms of air-reversed Brayton heat pump with a turbocharger describes the air state and thermodynamic process more intuitively and vividly.

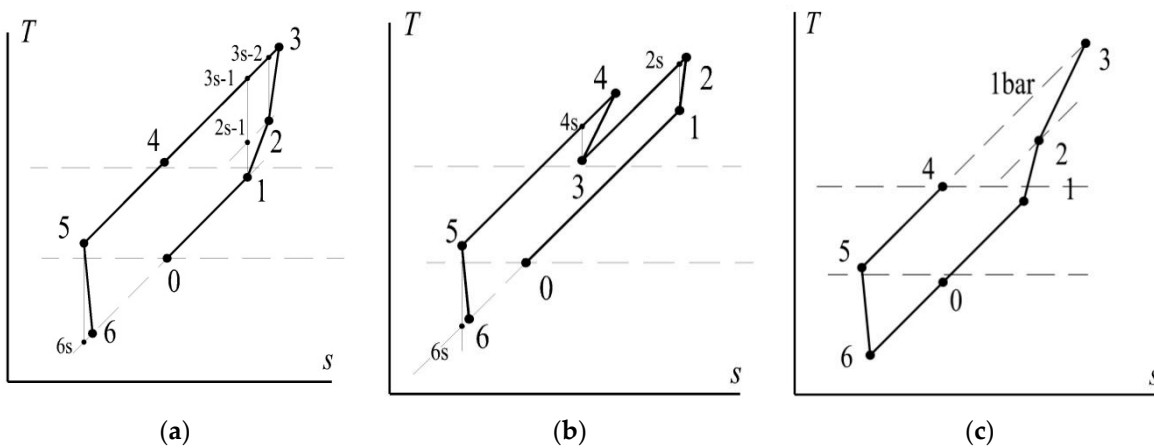

(a)                              (b)                              (c)

**Figure 2.** *Cont.*

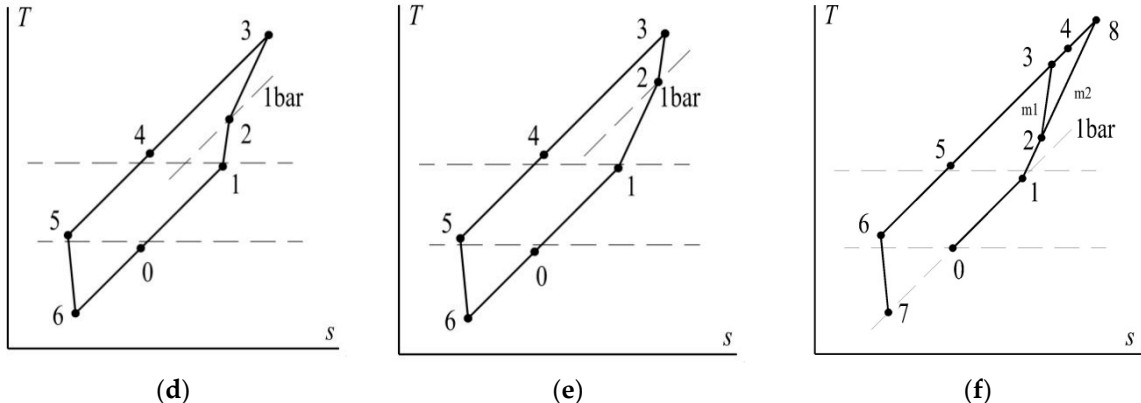

**Figure 2.** *T-s* diagram of different air-reversed Brayton heat pump systems with a turbocharger. (**a**) Cycle A; (**b**) Cycle B; (**c**) Cycle C; (**d**) Cycle D; (**e**) Cycle E; (**f**) Cycle F.

2.2.2. Mathematical Modeling

The mathematical model of Cycle A can be found in study [24]. It can be gathered from the system structure that the heating performance of Cycle B is slightly worse than that of other cycles. Its mathematical model is not described in this paper. The mathematical models of the other cycles are discussed where each cycle's hypothesis refers to Li et al. [24,25]. The mathematical model of Cycle C can be established by referring to the above thermodynamic model and *T-s* diagram.

The heating capacity of Cycle C is defined as:

$$Q_H = m_1(h_3 - h_4) = m_1 c_p(T_3 - T_4) \tag{1}$$

while the blower energy consumption and heating COP are given as follows:

$$W_f = m_1 c_p(T_3 - T_2) \tag{2}$$

$$COP_H = \frac{Q_H}{W_f} = \frac{T_3 - T_4}{T_3 - T_2} \tag{3}$$

The ideal gas equation is as follows:

$$pv = RT \tag{4}$$

The adiabatic process of the blower from 2–3 is described by the following expressions:

$$T_{3s} = T_2 \left(\frac{p_3}{p_2}\right)^{\frac{k-1}{k}} = T_2 \gamma_{2-3} \tag{5}$$

$$T_3 - T_2 = \frac{T_{3s} - T_2}{\eta_f} \tag{6}$$

The turbine polytropic process can be written as:

$$T_5 = T_{6s} \left(\frac{p_5}{p_6}\right)^{\frac{k-1}{k}} = T_{6s} \gamma_{5-6} \tag{7}$$

The compressor variable process is:

$$T_{2s} = T_1 \left(\frac{p_2}{p_1}\right)^{\frac{k-1}{k}} = T_1 \gamma_{1-2} \tag{8}$$

$$T_2 - T_1 = \frac{T_{2s} - T_1}{\eta_c} \tag{9}$$

The following expression gives the turbine efficiency equation:

$$T_5 - T_6 = (T_5 - T_{6s})\eta_e \tag{10}$$

For a better representation, the following one dimensionless parameter is defined:

$$\theta = \frac{T_4}{T_0} \tag{11}$$

According to the balance equation of the turbocharger, the following hold:

$$T_5 - T_6 = T_2 - T_1 \tag{12}$$

$$T_5\left(1 - \frac{1}{\gamma_{1-3}}\right)\eta = T_1(\gamma_{1-2} - 1) \tag{13}$$

$$\gamma_{1-2} = \frac{b\left(1 - \frac{1}{\gamma_{1-3}}\right)\eta}{a} + 1 \tag{14}$$

$$\gamma_{2-3} = \frac{\gamma_{1-3}}{\gamma_{1-2}} \tag{15}$$

where $a = 1 - \varepsilon_r + \varepsilon_r\theta$; $b = \theta - \varepsilon_r\theta + \varepsilon_r$; $\eta = \eta_c\eta_e$.

The balance equation of the regenerator is given as:

$$T_4 - T_5 = T_1 - T_0 = \varepsilon_r(T_4 - T_0) \tag{16}$$

$$T_1 = aT_0 \tag{17}$$

$$T_5 = bT_0 \tag{18}$$

and the heating COP as:

$$\begin{aligned}
\text{COP}_H &= \frac{Q_H}{W_f} = \frac{T_3 - T_4}{T_3 - T_2} = \frac{T_2\left(1 + \frac{\gamma_{2-3}-1}{\eta_f}\right) - T_4}{T_2\left(1 + \frac{\gamma_{2-3}-1}{\eta_f}\right) - T_2} = \frac{\left(1 + \frac{\gamma_{2-3}-1}{\eta_f}\right) - \frac{T_4}{T_2}}{\left(1 + \frac{\gamma_{2-3}-1}{\eta_f}\right) - 1} \\
&= \frac{\left(1 + \frac{\gamma_{2-3}-1}{\eta_f}\right) - \frac{\theta}{a\left(1 + \frac{\gamma_{1-2}-1}{\eta_c}\right)}}{\frac{\gamma_{2-3}-1}{\eta_f}}
\end{aligned} \tag{19}$$

The variable $\gamma_{2-3}$ is a function of $\gamma_{1-3}$. Therefore, $\text{COP}_H$ can be described as a function of the pressure ratio.

Cycle E is a closed system, which is conducive to the stability of the lubricating oil system and is not affected by environmental conditions. It is easy to exchange the cooling and heating functions, and it also has cooling and heating functions. Therefore, the cooling coefficient energy efficiency ratio (EER) + 1 is used to obtain the analytical expression of COP in the calculations, and the heat transfer temperature difference is used to correct the heat source temperature and heat sink temperature.

The cooling capacity can be written as:

$$Q_C = m(h_0 - h_6) = c_p m(T_0 - T_6) \tag{20}$$

The blower energy consumption and cooling *EER* are described by the following expressions:

$$W_f = mc_p(T_2 - T_1) \tag{21}$$

$$EER = \frac{Q_C}{W_f} = \frac{T_0 - T_6}{T_2 - T_1} \tag{22}$$

The compressor and blower isentropic compression processes are described by the following equation:

$$T_{3s-1} = T_1 \gamma_{5-6} \tag{23}$$

while the turbine expansion process is:

$$T_5 = T_{6s} \left( \frac{p_5}{p_6} \right)^{\frac{k-1}{k}} = T_{6s} \gamma_{5-6} \tag{24}$$

The balance equation of turbocharger can be written as follows:

$$T_5 - T_6 = T_3 - T_2 \tag{25}$$

$$(T_5 - T_{6s})\eta = T_{3s-2} - T_2 \tag{26}$$

According to the approximate parallel hypothesis, we obtain:

$$T_{3s-2} - T_2 = T_{3s-1} - T_{2s-1} \tag{27}$$

$$(T_5 - T_{6s})\eta = T_{3s-1} - T_{2s-1} \tag{28}$$

$$\gamma_{1-2} = \gamma_{5-6} - \frac{b}{a}\eta \left( 1 - \frac{1}{\gamma_{5-6}} \right) \tag{29}$$

The heat transfer process of the recovery heat exchanger can be written as follows:

$$T_1 - T_0 = T_4 - T_5 = \varepsilon_r (T_4 - T_0) \tag{30}$$

We define the following dimensionless parameter:

$$\theta = \frac{T_4}{T_0} \tag{31}$$

where

$$T_2 = T_1 + \frac{1}{\eta_f}(T_{2s-1} - T_1) \tag{32}$$

$$T_3 = T_2 + \frac{1}{\eta_c}(T_{3s-2} - T_2) \tag{33}$$

The *EER* can be written as only related to the turbine pressure and $\gamma_{5-6}$ as follows:

$$EER = \frac{T_0 - T_6}{T_2 - T_1} = \frac{1 - b\left[1 - \left(1 - \frac{1}{\gamma_{5-6}}\right)\eta_e\right]}{\frac{1}{\eta_f}a(\gamma_{1-2} - 1)} \tag{34}$$

$$COP = EER + 1 \tag{35}$$

The flow ratio of two blowers should be introduced as another variable for Cycle F with two parallel blowers. However, the mass flow rate is only related to the performance of the blowers; therefore, it is treated as a constant in the derivation of the expressions. The absolute performance of the system is lower than that of the other single-blower cycles; however, it is found through measurements that this cycle can ensure the efficient operation of the turbocharger, which is more in line with the actual operating conditions of the turbocharger and obtains a better practical, operational effect [26]. The specific calculation process is as follows.

The heating capacity is calculated as follows:

$$Q_H = (m_2 + m_1)(h_4 - h_5) = c_p(m_2 + m_1)(T_4 - T_5) \tag{36}$$

where $m_1$ and $m_2$ are parallel mass flows in Figure 2f, and the blower energy consumption is:

$$W_f = m_2 c_p (T_8 - T_1) + m_1 c_p (T_2 - T_1) \tag{37}$$

Assuming that the flow ratio $G$ is a known quantity, i.e.,

$$G = \frac{m_1}{m_2} \tag{38}$$

we can describe the heating COP as follows:

$$\text{COP}_H = \frac{Q_H}{W_f} = \frac{(m_2 + m_1) c_p (T_4 - T_5)}{m_2 c_p (T_8 - T_1) + m_1 c_p (T_2 - T_1)} \tag{39}$$

The variable process of two parallel blowers can be written as:

$$T_{2s} = T_1 \left( \frac{p_2}{p_1} \right)^{\frac{k-1}{k}} = T_1 \gamma_{1-2} \tag{40}$$

$$T_2 - T_1 = \frac{T_{2s} - T_1}{\eta_f} \tag{41}$$

$$T_{8s} = T_1 \left( \frac{p_8}{p_1} \right)^{\frac{k-1}{k}} = T_1 \gamma_{6-7} \tag{42}$$

$$\begin{aligned} T_8 &= T_1 + \frac{T_1 \gamma_{6-7} - T_1}{\eta_f} = T_1 \left( 1 + \frac{\gamma_{6-7} - 1}{\eta_f} \right) \\ &= T_0 (\theta \varepsilon_r - \varepsilon_r + 1) \left( 1 + \frac{\gamma_{6-7} - 1}{\eta_f} \right) \end{aligned} \tag{43}$$

and the heat transfer process of the recovery heat exchanger is described as follows:

$$T_5 - T_6 = T_1 - T_0 = \varepsilon_r (T_5 - T_0) \tag{44}$$

$$T_6 = T_0 (\theta - \theta \varepsilon_r + \varepsilon_r) = b T_0 \tag{45}$$

$$T_1 = T_0 (\theta \varepsilon_r - \varepsilon_r + 1) = a T_0 \tag{46}$$

The compressor variable process can be written as:

$$T_{3s} = T_2 \left( \frac{p_3}{p_2} \right)^{\frac{k-1}{k}} = T_2 \gamma_{2-3} \tag{47}$$

$$T_6 = T_{7s} \gamma_{6-7} \tag{48}$$

while the parallel blowers mixing process is:

$$T_4 = \frac{m_1 T_3 + m_2 T_8}{m_1 + m_2} = \frac{G T_3 + T_8}{G + 1} \tag{49}$$

We define the following one dimensionless parameter:

$$T_5 = \theta T_0 \tag{50}$$

The balance equation of turbocharger is given as follows:

$$(m_1 + m_2)(T_6 - T_7) = m_1 (T_3 - T_2) \tag{51}$$

$$(m_1 + m_2)(T_6 - T_{7s}) \eta_e = m_1 \frac{T_{3s} - T_2}{\eta_c} \tag{52}$$

$$(G+1)(T_6 - T_{7s})\eta_e = G\frac{T_{3s} - T_2}{\eta_c} \tag{53}$$

$$b(G+1)(1 - \frac{1}{\gamma_{6-7}})\eta = aG(1 + \frac{\gamma_{1-2} - 1}{\eta_f})(\gamma_{2-3} - 1) \tag{54}$$

Based on these assumptions, the following equations can be derived:

$$T_1\gamma_{6-7} - T_1\gamma_{1-2} = T_2\gamma_{2-3} - T_2 \tag{55}$$

$$(G+1)(T_6 - T_{7s})\eta = G(T_1\gamma_{6-7} - T_1\gamma_{1-2}) \tag{56}$$

$$\gamma_{6-7} = \gamma_{1-2}\gamma_{2-3} \tag{57}$$

$$\frac{b}{a}\left(\frac{G+1}{G}\right)\left(1 - \frac{1}{\gamma_{6-7}}\right)\eta = \gamma_{6-7} - \gamma_{1-2} \tag{58}$$

$$\gamma_{1-2} = \gamma_{6-7} - \frac{b}{a}\left(\frac{G+1}{G}\right)\left(1 - \frac{1}{\gamma_{6-7}}\right)\eta \tag{59}$$

$$\gamma_{2-3} = \frac{\gamma_{6-7}}{\gamma_{1-2}} \tag{60}$$

Based on the above expression, the calculation method of $COP_H$ related to $\gamma_{6-7}$ can be derived as follows:

$$COP_H = \frac{Q_H}{W_f} = \frac{(m_2 + m_1)c_p(T_4 - T_5)}{m_2 c_p(T_8 - T_1) + m_1 c_p(T_2 - T_1)} = \frac{(1+G)c_p(T_4 - T_5)}{c_p(T_8 - T_1) + Gc_p(T_2 - T_1)}$$

$$= \frac{(1+G)\left(\frac{Ga(1 + \frac{\gamma_{1-2} - 1}{\eta_f})\gamma_{2-3} + a(1 + \frac{\gamma_{6-7} - 1}{\eta_f})}{G+1} - \theta\right)}{a\frac{\gamma_{6-7} - 1}{\eta_f} + aG\frac{\gamma_{1-2} - 1}{\eta_f}} \tag{61}$$

## 3. Results and Analysis

### 3.1. Theoretical Calculation Results

Using the above mathematical models to calculate the heating COP, the following results were obtained according to the calculation results shown in Figures 3 and 4. It can be noted that the heating performance of Cycle C and that of Cycle A were similar. They were both the highest, followed by Cycles E, B, and F under all working conditions.

There was an optimal pressure ratio for all cycles under certain working conditions. As the pressure ratio gradually reached the optimal value, the heating performance also reached a peak value (the optimal COP). Afterwards, the heating COP could not increase with an increase in the pressure ratio. When the ambient temperature was fixed at −15 °C at the heating water temperature of 45 °C, the optimal pressure ratio was about 1.5–1.7 for each single-blower cycle. The optimal pressure ratio of the double-blower cycle was close to 1.8–1.9, and the optimal COP ranged from 1.22 to 1.27. When the water supply temperature dropped to 35 °C, the optimal pressure ratio of all cycles remained almost unchanged, with the optimal COP in the range of 1.24–1.29. Therefore, the heating was not influenced by the water temperature at a fixed ambient temperature. When the water supply temperature was fixed, the optimal pressure ratio of each cycle showed a minor decrease as the ambient temperature rose from −15 °C to 5 °C. It should be noted that the optimal pressure ratio was not sensitive to the ambient temperature. Therefore, for the air-reversed Brayton heat pump systems with a turbocharger driven by blowers, the ambient temperature and the water supply temperature were not the main factors that affected the optimal pressure ratio.

Figures 3 and 4 show that once the pressure ratio between the turbine inlet pressure and the outlet pressure reaches a certain value, the COP of heating did not increase with an increase in the pressure ratio. Therefore, for the reversed Brayton heat pump with a single blower, this heat pump maintained a high heating efficiency when the turbine pressure ratio

was greater than 1.5. For Cycle C, when the water supply temperature varied from 45 °C to 35 °C, the optimal COP increased by about 0.04 under the same ambient temperature. The ambient temperature varied from −15 °C to 5 °C, and the optimal COP rose by about 0.09 at the same water temperature. Therefore, the COP in the optimal heating system was insensitive to the ambient temperature and the temperature of hot water, which is also a significant advantage.

The calculation results also showed that the highest COP of the single-fan heat pump system under different working conditions was maintained in the range of 1.2–1.4. Taking the example of Cycle A, it can be observed that the COP increased by at least 27% while the turbocharger compressor efficiency and the turbine efficiency increased to 0.85 and 0.9, respectively [27]. Apparently, the heating performance of the system is significantly improved as the efficiency of the turbocharger is improved. Therefore, it requires more attention. In addition, the variation range of air refrigerant pressure was not large under the operating conditions, which reduced the requirements of equipment airtightness. Consequently, it reduced the processing difficulty.

*3.2. Experimental Results*

The above theoretical results pointed to the way to establish the test bench and provide support for the late promotion. The relevant experimental rigs are shown in Figure 5, which integrate the traditional, regenerated air-reversed Brayton heat pump cycle and the cycles with a turbocharger and blowers, as described in Section 2.1. Based on a reconstruction of this test bench, more cycles under different working conditions were realized. The experimental conditions and specific results were provided in the literature by Li et al. [26].

The representative Cycle F, which was simple to implement in the existing experimental bench, was selected as the typical one for comparison in this paper. Figure 6 shows the theoretical and measured results of Cycle F under the same initial conditions. Here, the calculated results were in good agreement with the experimental observation, and the maximum error was lower than 5.6%. Because of the assumption that the pressure lines are approximately parallel within a certain range of entropy in-crease in the derivation of the formula [24], but the measured system energy consumption is close to a fixed value, due to the use of a constant-frequency blower as the power equipment in actual measurement it was different from the power calculated by the enthalpy difference in the calculation. So, there were some deviations between the calculated and experimental results. Moreover, the computation procedure neglected a few factors, such as heat transfer from the heat pump system to the outside, friction resistance in the pipe, and so on. In general, the expressions can be used to predict the ranges of turbine pressure value and COP, which can simplify the calculation process, optimize the system form, and achieve the optimal pressure ratio quickly when designing experiments or products.

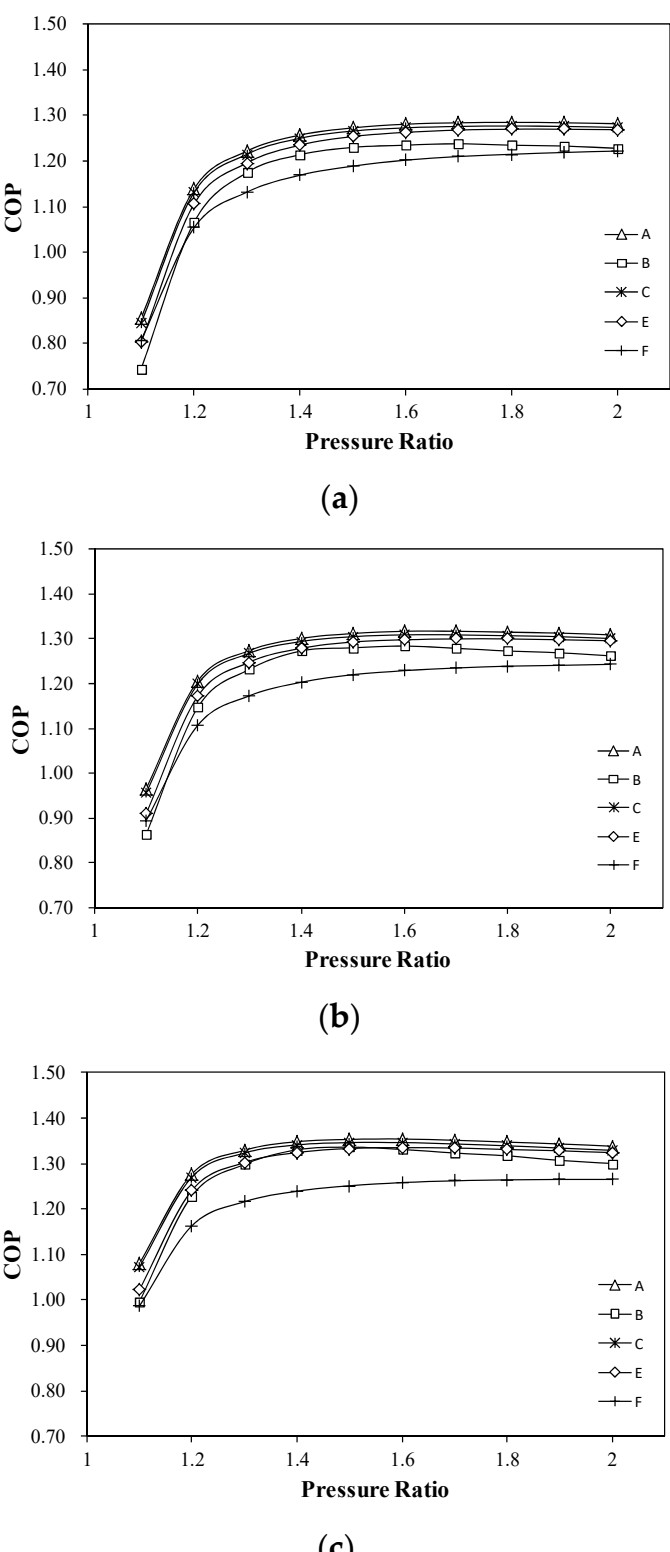

**Figure 3.** Variation of COP versus pressure ratio at 45 °C hot water temperature. (**a**) Ambient temperature of −15 °C. (**b**) Ambient temperature of −5 °C. (**c**) Ambient temperature of 5 °C.

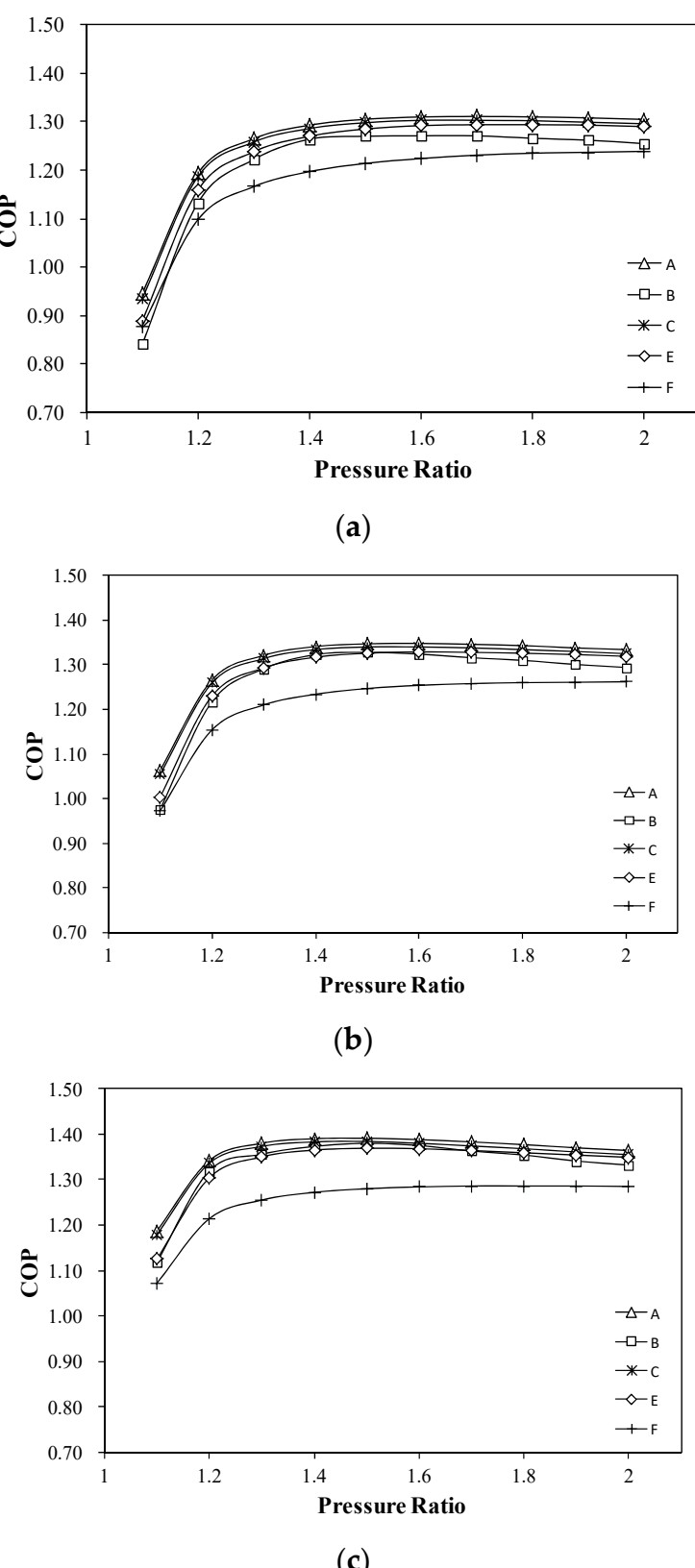

**Figure 4.** Variation of COP versus pressure ratio at 35 °C hot water temperature. (**a**) Ambient temperature of −15 °C. (**b**) Ambient temperature of −5 °C. (**c**) Ambient temperature of 5 °C.

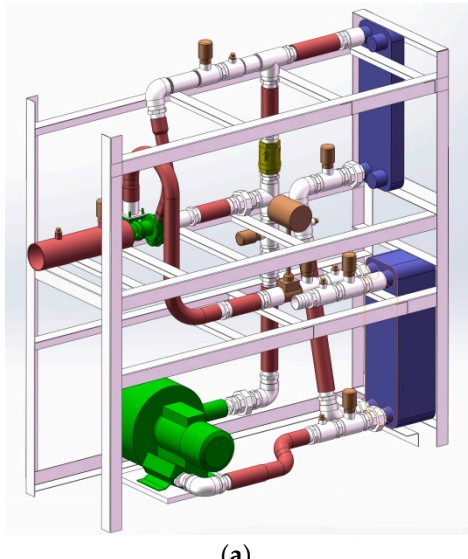

(**a**)

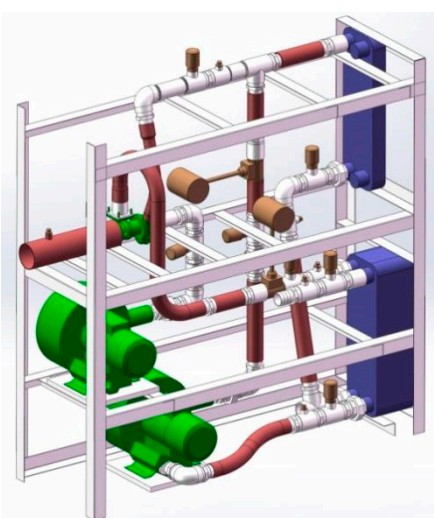

(**b**)

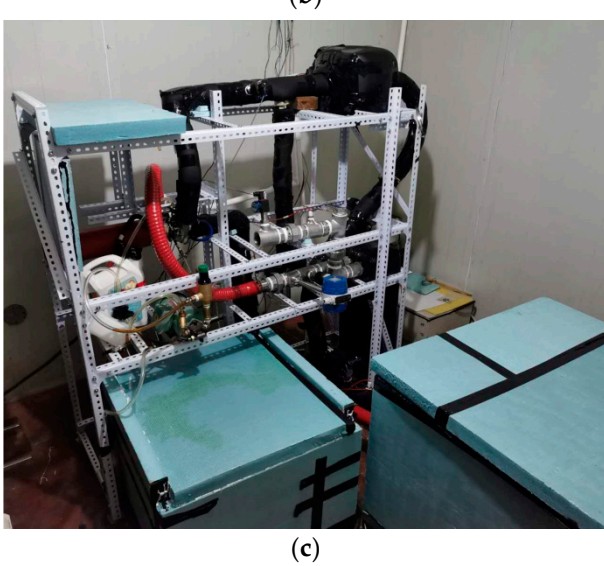

(**c**)

**Figure 5.** Diagrams of the test bench of the air-reversed Brayton heat pump with a turbocharger.
(**a**) One-blower system. (**b**) Two-blower system. (**c**) Photograph of the air cycle heat pump system.

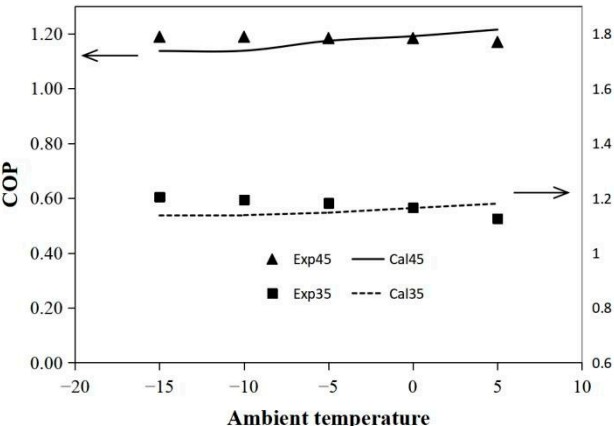

**Figure 6.** Comparison of calculation and experimental results under different conditions (45 °C/35 °C hot water temperature).

## 4. Conclusions

Six types of air-reversed Brayton heat pump system with turbochargers driven by blowers were proposed according to the location of the system in contact with the environment and the number and position of the blowers. The thermodynamic expressions of the heating COP and the corresponding turbine pressure ratio were derived and analyzed, respectively. The main conclusions were drawn as follows:

There is an optimal pressure ratio for each cycle. The optimal pressure ratio of the single-blower system is about 1.5–1.7, while that of the dual-blower system is about 1.8–1.9.

Cycle A (with a single blower before the compressor, open to the heat source side) and Cycle C (with a single blower after the compressor, open to the heat sink side) have the highest heating COPs and are worthy of more attention.

The theoretical results of Cycle F were in agreement with the experimental observations, with a maximum error of less than 5.6%.

## 5. Prospects

In this paper, the air-reversed Brayton heat pump was mainly used for a heating system. However, it can be applied under circumstances with a large temperature difference between the heat source and the sink, such as high-temperature water supply or cryogenic refrigeration, high-temperature drying, and combined heating with other heat pumps which meet the demands of different heating conditions and building functions. All these issues need to be further studied. In addition, it is necessary to consider the dynamic coupling characteristics between the building thermal environment and the air-reversed Brayton heat pump system, as well as the association between the energy consumption of the system and the building and climatic conditions throughout a whole year.

**Author Contributions:** Conceptualization, S.W.; methodology, S.W. and S.L.; investigation, S.L. and S.W.; writing—original draft preparation, S.L. and S.J.; writing—review and editing, S.W., S.L., S.J. and X.W. All authors have read and agreed to the published version of the manuscript.

**Funding:** This research received the support from the PhD Start-up Fund of the Natural Science Foundation of Liaoning Province, China (grant number 2019-BS-055).

**Institutional Review Board Statement:** Not applicable.

**Informed Consent Statement:** Not applicable.

**Data Availability Statement:** The data presented in this study are available on request from the corresponding author.

**Conflicts of Interest:** The authors declare no conflict of interest.

## Nomenclature

| | | | |
|---|---|---|---|
| Nomenclature | | | |
| $c_p$ | specific heat at constant pressure (J/kgK) | COP/EER | coefficient of performance |
| $h$ | enthalpy (J/kg) | $Q_H$ | heating capacity (W) |
| $m$ | mass flow rate (kg/s) | $Q_C$ | cooling capacity (W) |
| $p$ | pressure (Pa) | $T$ | temperature (°C/K) |
| $R$ | gas constant (J/Kmol) | $v$ | specific volume (m$^3$/kg) |
| $W_f$ | blower energy consumption(W) | | |
| Greek symbols | | Subscripts | |
| $\eta$ | effectiveness | c | compressor |
| $\theta$ | temperature ratio defined in Equation (11) | f | blower |
| $\varepsilon_r$ | effectiveness of regenerator | e | expander; turbine |
| $\gamma$ | a function of pressure ratio | s | isentropic |

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
