# Peer review of "Analysis of the Air-Reversed Brayton Heat Pump with Different Layouts of Turbochargers for Space Heating"

_buildings, doi:10.3390/buildings12070870_

Round 1

Reviewer 1 Report

The paper describes the behavior of 6 types of Brighton circuits. The authors have clearly presented the calculation methodology to determine the COP at different bottom and top heat source temperatures. The results presented in the form of graphs show the correct behavior of the circuits. One puzzling thing is Figure 6, where the behavior of the COP depending on the outside temperature is not entirely clear. According to my knowledge with the increase of the outside temperature COP should be increasing, but the experimental data do not confirm this, especially at temperature of 35 degrees C, where the graph is even falling. I am curious what could be the reason for this, which should be explained. In the conclusion section it was shown that for a specific circuit there is an optimum value of pressure ratio at which the circuit reaches its maximum concerning COP, which was the thesis of the research undertaken.

Reviewer 2 Report

In this article, six types of air reversed Brayton heat pump systems with turbochargers driven by blowers were proposed for different working environments and applications. Some comments were raised after the review of this manuscript.

Comments:

1.      The abstract is not quite informative. It is better to present more specific data or results.

2.      Literature should be critically reviewed, and novelty should be clearly justified and presented.

3.      For the analysis in section 2.2.1, the authors presented many equations. What’s new? Actually, it is easy to analyze these.

4.      The equations were not numbered.

5.      For the important part (3.2. Experimental results), the important results of COPs of the proposed cycles were not presented and compared.

6.      The conclusions are not concise.

7.      Unit formats are not consistent throughout the whole manuscript, i.e., oC/K, J kg-1, kg s-1, etc.

8.      References were not well formatted, i.e., Zhang et al [13], etc.

Round 2

Reviewer 2 Report

Some minor comments:

1.      The equations were not numbered.

2.      The conclusions are not concise.
